# Biocapacity—Premise of Sustainable Development in the European Space

**Rodica-Manuela Gogonea [1,\*], Simona Ioana Ghita [1,2] and Andreea Simona Saseanu [3]**

[1]  Department of Statistics and Econometrics, Bucharest University of Economic Studies, 010552 Bucharest, Romania; simona.ghita@csie.ase.ro

[2]  Institute of National Economy, 050711 Bucharest, Romania

[3]  Department of Business, Consumer Sciences and Quality Management, Bucharest University of Economic Studies, 010374 Bucharest, Romania; andreea.saseanu@com.ase.ro

\*  Correspondence: manuela.gogonea@gmail.com or manuela.gogonea@csie.ase.ro; Tel.: +40-724-742-477

**Abstract:** The contemporary lifestyle, based on unsustainable consumption patterns, leads to an orientation of the society towards the development and application of sustainable development strategies and policies. The comparative analysis of the ecological footprint and biocapacity allows one to study the interaction between human activities and environment, through the biocapacity reserve or deficit. In this context, this article carries out a complex analysis of the biocapacity reserve/deficit, as a latent variable that quantifies sustainability, viewed through a selection of determinants, from which three main components have been extracted: A component of education and social exclusion, a component of economic development, innovation, and environment, and a demographic component. These were transformed—through a multiple linear regression model—into exogenous variables with high explanatory power over the variation of the biocapacity reserve/deficit and constituted the tools in identifying behavioral patterns of the European countries and a set of measures leading to the sustainability of the ecological reserve.

**Keywords:** ecological footprint; biocapacity; biocapacity reserve/deficit; sustainability

---

## 1. Introduction

In recent decades, the pacing of environmental destruction has intensified considerably, bringing the day when human resource consumption exceeds the resource regeneration capacity closer and closer every year ("Earth Overshoot Day"). According to the Global Footprint Network, this day has been offset by two months in the last two decades [1]. The current pace of economic development has led to an increase in the incidence and frequency of extreme climatic phenomena, which comes as a negative reaction of the environment, in response to the increasing pressure imposed by human activities. Maintaining a balance between consumption and the availability of resources has become a real challenge [2]. That is why numerous studies in the specialized literature have focused on the analysis of variables that reflect the environment state and its interaction with human activities: Ecological footprint and biocapacity.

According to the Global Footprint Network, ecological footprint can be defined as: „A measure of how much area of biologically productive land and water an individual, population or activity requires to produce all the resources it consumes and to absorb the waste it generates, using prevailing technology and resource management practices", while—according to the same source—biological capacity or biocapacity constitutes "The capacity of ecosystems to regenerate what people demand from those surfaces" [3]. The main variable that reflects the interaction between the two parts, analyzed in detail in this article, is the biocapacity reserve or deficit, which represents the difference between

biocapacity and ecological footprint of a region or country. If biocapacity is larger than ecological footprint, then there is a biocapacity reserve. In the opposite situation, there is a biocapacity deficit. All these indicators are usually expressed in global hectares, but—to ensure comparability—the ratio between each indicator and population size, measured in global hectares per capita, was used in the analysis.

The main objective of the present research is to carry out a complex, multidimensional analysis of the biocapacity reserve/deficit, in order to identify its main determinants, covering all three main dimensions of sustainable development: The economic, social, and environmental pillar, as well as highlighting the tools through which one can intervene to restore a balance between consumption and availability of resources.

The research is structured in six sections. Section 1 (Introduction) presents the context, the need, and the importance of the topic addressed in the article. It continues with a detailed analysis of the results of the main studies in the field, existing in the specialized literature, highlighting the different points of view of the specialists on the sustainability of the economic activity and environment, in Section 2 (Literature review). Section 3 (Materials and Methods) contains a description of the statistical and econometric methods, as well as the set of statistical variables used in studying the biocapacity reserve/deficit. The analysis included in Section 4 aims at identifying the main components that determine the biocapacity reserve/deficit, measuring their influence, and characterizing behavioral patterns of the European countries with respect to the biocapacity reserve/deficit and its determinants. "Discussions" and "Conclusions" sections (Sections 5 and 6) summarize the main results of the study and their importance, presenting the main economic, social, and environmental policy measures that can be applied to mitigate the imbalances between resource consumption and regeneration, to reduce the biocapacity deficit according to the behavioral patterns of European countries.

## 2. Literature review

Our planet with its resources is considered, in economic theory, as one of the production factors without which the economic activity could not be carried on. Nowadays, we are living and developing from the resources that should have been available to future generations, endangering their chances of living and developing properly. Moreover, not all nations of the world benefit from the resources of the planet equally, but there is also an uneven distribution of both resources and consumption, with some countries developing their economy on account of the natural resources of other countries, limiting them. Scientists have studied and proposed several ways to measure the needed and consumed resources in the production process, but also the impact of consumption on the environment. One of the notable contributions to quantifying the pressure that human activity puts on the environment belongs to Mathis Wackernagel and William Rees who, in 1994, introduced the concepts of "ecological footprint" and "biocapacity" (or ecological capacity) [4]. The last one is considered to be an estimation of the ecosystem's production of various biological materials (e.g., natural resources) to which both absorption and filtration from other materials (e.g., carbon dioxide in the atmosphere) are added [5]. Ecological footprint and biocapacity are integrated into the global footprint indicator system, allowing a comparative analysis that generated significant contributions in understanding the manifestation of sustainability phenomenon [6].

Thus, the difference between ecological footprint and ecological capacity represents the biocapacity reserve or deficit—concepts that Wackernagel and Rees also contributed to [7]. The biocapacity deficit is considered in most situations as a result of human transgression, of climate change that occurs mainly as a result of unrestricted $CO_2$ atmospheric emissions [8,9]. From a comparative perspective, the countries of the world are classified as economic debtors, when ecological footprint exceeds biocapacity and economic creditors in the opposite situation when ecological footprint is smaller compared to biocapacity [10–12].

Numerous studies have considered that the indicator system of ecological footprint, which also includes biocapacity reserve/deficit can be used in analyzing the relationship between human activities

and the environment. Considering the purpose of the present paper, the analysis of the specialized studies is focused on the following aspects: The categories of main drivers of ecological footprint (or of other indicators in the ecological footprint system) that have been identified in various specialized studies; typology of determinants' impacts or influences on biocapacity; and methods of identifying drivers and quantifying their influences.

Among the determinants of biocapacity or ecological footprint, a first category of studies identifies factors with high potential influence: Demographic factor, level of economic development, urbanization rate, and industrialization level.

Thus, Jia et al. [13] analyzed the biocapacity deficit in the Henan region of China—on the background of an accelerated growth of the ecological footprint and a low biocapacity level. The authors identified the main drivers with significant explanatory power of the variability in the ecological footprint: Demographic factor (population), economic development factor (GDP per capita), and urbanization factor (the share of urban population), all of these posing an excessive pressure on the environment, leading to an increase in the ecological footprint. Tang, Zhong, and Liu (2011) [14] used a model that allows decomposing the impact of human activity on the environment. The results showed that the population is the main factor that explains the variation of the ecological footprint, while the industrialization and urbanization are positively correlated with the ecological footprint. Also, the conclusions of the studies show that starting from the same set of hypothetical determinants of the ecological footprint or biocapacity, the results may differ depending on the method used.

Similar results expressing the negative impact of demographic factor and urbanization on environmental degradation were also observed in the studies of Pacheco et al. (2014) and Valle Junior et al. (2014) [15,16]. According to these studies, the agriculture and urbanization process have a negative impact on the environment at different ecosystem levels, signaling a degradation and depletion problem of natural resources, as a result of the increase in the pollution level and the necessity of improving the economic systems worldwide. Numerous studies have shown that the ability of the planet to support the population is significantly influenced on the one hand by the resource consumption and by the waste production rates, and on the other hand by the living behavior, structure, institutions, and production technology, which are in an accelerated dynamic process [17–19]. At the same time, the high ecosystem conversion degree, with an emphasis on agriculture and urbanization, is an important factor of the biodiversity variations [20,21].

Another category of studies analyzes the effect of the research innovation activity on the eco-friendly behavior of the companies and on the environment quality. The results show that—in the long and short term—the effects of innovation on environmental indicators can be positive or negative.

Technological progress and innovation, in general, have a great contribution in optimizing the manufacturing process, in emphasizing its ecological character, by reducing the consumption of materials and pollution, by increasing the use of "green", renewable energy sources, by obtaining new biodegradable materials, thus helping to relieve the pressure of economic activities on the environment and to mitigate climate change [22–26]. In this regard, one of the European Union's actions towards limiting climate change and reducing greenhouse gas emissions was the launch of the EU Emissions Trading System (EU ETS) in 2005.

A study of the European Commission on this topic [27] analyzes the environmental impact of innovative bio-based products in order to support the implementation of the EU Plastics Strategy. The results of the study show that for bio-based products, the most important environmental impact concerns climate change, depletion of non-renewable energy resources, and human health, the three aspects cumulating between 30% and 60% of the total impact. Gagelmann and Frondel [28] analyzes the impact of EU ETS on innovation reflected in specialized studies, showing that the innovation effect is initially limited, but is followed by the adoption of numerous cost reduction strategies. Most of these theoretical studies do not reach a common conclusion on the increased impact of EC ETS on innovation, compared to other existing policy instruments. Anderson, B., Convery, F., Di Maria, C., [29] study the effect of implementing the EU ETS [30]—pilot phase—on the behavior of Irish firms, regarding

the adoption of new CO2 low-level technologies, as a consequence of changes in the emissions price. The survey results showed that this measure led to the changes in the companies' strategies, with almost half of the investigated companies reporting the adoption of new environmentally friendly technologies. Such changes in companies' behavior may have a positive influence on the environment quality.

Ellerman and Buchner, Grubb, Azar and Persson, Jaffe, Newell and Stavins [31–33] also brought other important contributions on the impact of adopting the EU ETS. A higher responsibility of the companies' activity in the direction of sustainability can improve their image and can increase the customer portfolio. The analysis of the companies' availability in adopting an eco-friendly behavior with an impact on reducing the carbon footprint can also be found in the studies of Porter and Kramer, Brewer, and De Marchi [34–36]. Thus, De Marchi [36] analyzes the relationship between the companies' cooperation availability in applying the research-development strategies and their inclination to introduce environmental innovations, based on the data obtained through "Community Innovation Survey on Spanish Manufacturing Firms—PITEC". The results show that companies adopting environment innovative strategies cooperate more with external partners than other innovative companies. Also, the presence of a substitution effect between the cooperation activity and the internal research-development effort of the companies is highlighted. Brunnermeier and Cohen [37] analyze the determinants of the environmental innovation process based on panel data models. For the quantification of environmental innovation, the number of patents and environmental innovations applied in the industry was used, the results showing that environmental innovation is significantly correlated with the increase of pollution reduction expenditures.

Some studies highlighted the existence of a spatial component in explaining the variations of the ecological footprint. Thus, Ramirez Y. N. [38] carried out an analysis of the link between the sustainability components and the ecological footprint, using a spatial regression model. Thus, he predicted the ecological footprint level based on three explanatory variables: The forest coverage, the literacy rate, and the aqueduct coverage, with these variables representing the environmental, the socio-human (educational), and the urbanization factor. The results showed that—in addition to the significant influence of the three mentioned variables—there is also a significant influence of the ecological footprint of the neighboring regions on the ecological footprint of the analyzed region, emphasizing that the economic relations between the regions imply an interdependence of their environmental variables as well.

In most of the studies presented above, the identification of the factors with significant influence on the environmental indicators, as well as the quantification of these influences, were achieved by using the econometric regression models. Other authors focus their analysis on composite indicator systems, which would surprise the multiple facets and complexity of the phenomenon studied.

Thus, Singh et al. [39] performs a presentation and a comparative analysis of the existing indicator systems, used in quantifying sustainability level and progress towards sustainable development, as well as the pressure imposed by human activities on the environment. He analyzes the quality of the composite indicators, referring to the three stages of their construction: Normalization, weighting, and aggregation. The efficiency of such indicators lies in the multilateral and integrative approach of the economic, social, and environmental aspects, in capturing the links between them and their dynamics. The indicators included in these systems have been classified into several groups, with ecological footprint being part of the ecosystem group [40]. Hoekstra and Wiedmann [41] analyze the existing footprints, comparing them with the maximum levels for ensuring sustainability. They underline the need for a footprint combination, for the improvement of the calculation techniques, for the high accuracy forecasting of the maximum sustainable levels, and the evaluation of the efficiency of the resource use.

A composite indicator is also used to measure the level and quality of human factor development: Human Development Index, with numerous studies highlighting the existence of a direct, positive link between the quality of the human factor and the level of the ecological footprint. The Human Development Index (HDI) combines three main components: The level of economic development (GDP

per capita), the state of health (life expectancy), and the educational level (Educational attainment). Starting with 2011, the classic version of HDI was replaced by an index that takes into account the inequality of income distribution, education, and health: Inequality-Adjusted HDI. Studies have shown that the direct correlation between the level of human development and the ecological footprint is maintained, with countries with high levels of human development having a higher life standard, but this is achieved through an increase in the level of ecological footprint [42].

The estimated amount of biocapacity is of great importance for understanding the implications on global sustainability, on national and international policies integrated into space management process. As a biosphere subsystem, the economy experiences a dynamic development, being increasingly difficult to identify its operation mode [43,44]. General policies have been developed gradually, targeting planetary boundaries through which the "safe storage space" for humanity has been defined [45,46]. In this context, significant for the analysis of sustainable regional economic systems is the inclusion of spatial analysis in biocapacity supply and demand [47]. Biocapacity concept represents—in the context of sustainable development—a landmark, on one side of social development, and on the other side a landmark of human welfare [48–51].

The countries' performance worldwide should be influenced in the global market by their different behaviors towards environment and sustainability problems, which will be the result of policies and strategies aimed at restoring the planet's resources.

## 3. Materials and Methods

The paper presents a multidimensional statistical-econometric analysis of the behavior of the environmental variables, especially of the reserve or deficit of biocapacity, identifying its main determinants, which characterize seven dimensions: Economic development, demography, education, health, social exclusion, research (innovation), and environment. The selection process of these dimensions started from the three pillars of sustainability, defined in the specialized literature: The economic pillar, the social pillar, and the environment pillar. The economic pillar was characterized through the macroeconomic output of human activity (which synthesizes the level of economic development of a country), but also through the research development and innovation activity, which numerous specialized studies indicate as having a special role in ensuring sustainable development. Furthermore, numerous reports capture the connection between the human factor and the environment status, which is why the social pillar was "broken" into four sub-pillars, in order to express the quantitative and qualitative sides of human development level. Thus, the magnitude of the demographic factor, the educational level of human factor, its state of health, and its poverty level were also taken into consideration.

The research was carried out on a set of 15 variables, outlining the following seven dimensions: Economic development, demography, education, health, social exclusion, research (innovation), and environment (Table 1).

The data refer to the European countries (EU or non-EU countries), for 2016 (the most recent year for which data on ecological footprint, biocapacity, and biocapacity reserve are available) and have been collected from the following sources:

- 2019 Edition National Footprint and Biocapacity Accounts (data year 2016) [52]—for data on ecological footprint, biocapacity, and biocapacity reserve;
- EUROSTAT [53]—for the other variables.

**Table 1.** The dataset.

| Pillar | Variable Name | Measurement Unit |
|---|---|---|
| Environment | Ecological footprint | Global hectares per capita |
| | Biocapacity | Global hectares per capita |
| | Biocapacity reserve/deficit (*Biocap_res*) | Global hectares per capita |
| | Impacts of extreme weather and climate related events-Direct Economic Losses (*Direct_loss*) | Euro per capita |
| Economic development | Real GDP per capita (*GDP_cap*) | $ per capita |
| Demography | Population density (*Pop_dens*) | Inhabitants per $km^2$ |
| Education | Participation rate in selected education levels-All ISCED 2011 levels excluding early childhood educational development (*Part_educ*) | % of total students in the age segment |
| | Expected school years of pupils and students-All ISCED 2011 levels excluding early childhood educational development (*School_years*) | Year number |
| | Share of 15-years-old students with low educational performance-Underachieving 15-year-old students by field-PISA survey (average of maths, reading and science score) (*Underachv*) | % of total students in the age segment |
| Health | Life expectancy at birth (*Life_exp*) | Years |
| | Health care expenditures (*Health_exp*) | % of GDP |
| Social exclusion | Material deprivation rate (*Mat_dep*) | % of total households |
| | In-work at-risk-of-poverty rate (*Work_pov*) | % of total households whose head is active on the labor market |
| Research-innovation | Number of researchers (*Researchers*) | % of total employees, expressed in full-time equivalent. |
| | Business expenditure on R&D (*RD_exp*) | % of GDP |

Source: authors' selection.

During the research, based on the necessity—suggested by Singh et al. 2009 that the indicator system has to ensure a multilateral and integrative approach to the three main pillars of sustainable development: Economic, social, and environmental pillar—the following hypotheses were formulated:

**Hypothesis 1:** *The level of economic development of a country has a significant influence on the behavior of the biocapacity reserve/deficit.*

**Hypothesis 2:** *Population density leads to significant differentiation in the biocapacity reserve/deficit.*

**Hypothesis 3:** *The degree of socio-human development (in terms of educational level and performance, health, and social exclusion) has a statistically significant influence on the behavior of the biocapacity reserve/deficit.*

**Hypothesis 4:** *The research innovation activity has a statistically significant impact on the variation of the biocapacity reserve/deficit.*

A large variety of statistical econometric methods was used in verifying these hypotheses, such as: Descriptive statistical analysis of frequency distributions, descriptive analysis methods of time and territorial statistical series, complex methods of multidimensional statistical analysis (hierarchical clustering methods, principal component analysis, regression, and correlation analysis).

Thus, in the first part of the research, a descriptive analysis of the ecological footprint and biocapacity was carried out for the European countries, accompanied by a cross-sectional and longitudinal analysis of the biocapacity reserve/deficit in this region. In the second part of the analysis, the main determinants with significant influence on the changes of the biocapacity reserve/deficit were identified, on the following dimensions: Economic development, demography, education, health, social exclusion, research innovation, and environment, which were included in a cluster analysis, in order to identify and characterize behavioral patterns of European countries depending on the magnitude of the biocapacity reserve/deficit and its determinants. However, the variables/factors used to explain the behavior of biocapacity reserve/deficit were numerous, which makes their generated effect difficult to interpret. Therefore, in the third part of the analysis, a reduction of the set of biocapacity reserve/deficit factors was performed, a reduction to a number of main components, by applying the principal component analysis. In the fourth part of the research, the influence of the main components previously identified on the variability in the biocapacity reserve/deficit was quantified. In the Conclusions section of the paper, the measures of economic, social and environmental policy were formulated in order to reduce the imbalances in biocapacity reserve, through the results of the multidimensional analysis carried out (Figure 1).

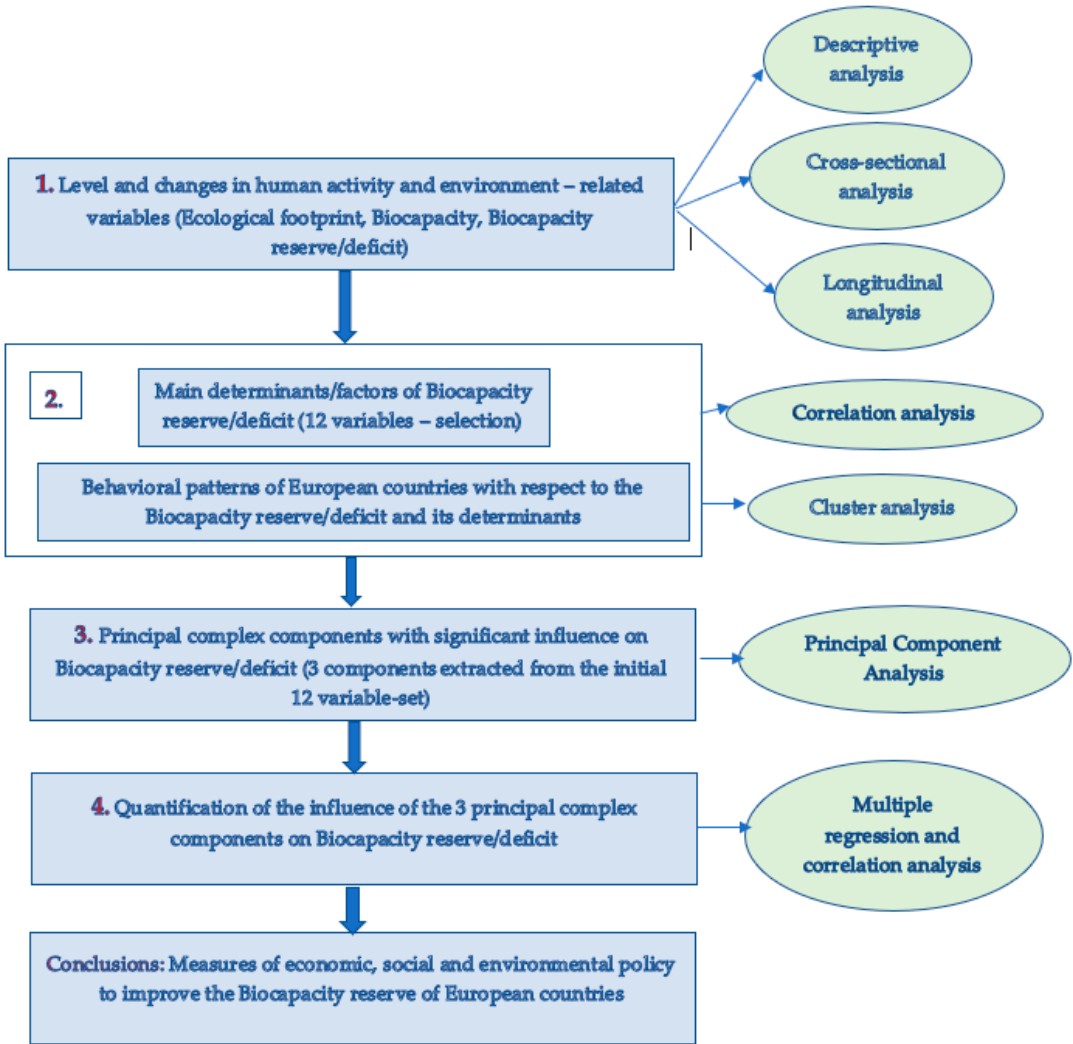

**Figure 1.** Analysis stages. Source: Authors' contribution.

The cluster analysis was applied for the purpose of grouping European countries according to a set of variables, so as to ensure homogeneity within groups and heterogeneity between groups. The set of grouping variables includes—besides the response variable (the biocapacity reserve/deficit)—a set of 12 factorial variables, which can influence the variation of the explained variable. Thus, the countries with the highest similarity of the grouping variables, which have a similar behavior in terms of biocapacity reserve and its determinants, were grouped in the same cluster, maintaining a higher dissimilarity between the groups of countries. Specifically, the Hierarchical Cluster Analysis was applied, using Ward's method of minimizing variability within groups. Through this method, a hierarchy of clusters was built, starting from the base level in which each country is constituted in an individual cluster; then, as it moves to the next level of the hierarchy, the pairs of clusters are merged into a new one (agglomerative clustering). The cluster merging criterion was the Euclidean distance between the pairs of observations. An optimal number of homogeneous clusters was established, and the grouping variables were standardized, in order to assure the data comparability. The countries in each cluster are characterized by a series of common features regarding the biocapacity reserve/deficit and its determinants, the method facilitating the identification of behavioral patterns of these countries from these points of view. By applying the Principal Component Analysis method, the set of variables used in the clustering of countries has been restricted, reduced to smaller dimensions, looking for the existence of "underlying variables" (factors) that explain some of the correlation between the initial variables. This way, the grouping of the multiple initial variables into a small number of components was performed, expressed as artificial variables that manage to retain an important part of the information of the initial dataset.

At the beginning part of the Principal Component Analysis, it is checked whether the initial set is suitable for applying this method, by verifying the following assumptions: The existence of continuous numerical variables; the existence of a linear relationship between variables; ensuring a minimum sample size; the adequacy of data for the application of the reduction procedure; verifying the existence of "outlier" values.

The 12 variables included in the analysis, which characterize the level of economic development, health, education, social exclusion, research innovation activity, as well as the interaction with the environment are continuous numerical variables, measured on a ratio scale. Each of the 12 variables has 30 values, corresponding to the 30 European countries included in the study. The relationships between these variables were analyzed using the Pearson linear correlation coefficient, resulting in the existence of significant linear correlations. The existence of sampling adequacy was verified with the help of the Kaiser-Meyer-Olkin indicator, and the appropriateness of applying the methods of reducing the size of the dataset was analyzed with the help of Bartlett's Test of Sphericity.

A value that exceeds the minimum level of 0.6 of the Kaiser-Meyer-Olkin (KMO) indicator ensures that the method is used properly. At the same time, the "Bartlett's Test of Sphericity" tests the null hypothesis that the correlation matrix is an identity-type matrix. If there is sufficient evidence, for 5% significance level to support the adoption of this hypothesis, then the Principal Component Analysis method should not be applied. In order for this method to be appropriate for the initial dataset, it is necessary to reject the null hypothesis. The existence of extreme values was signaled with the help of the SPSS program, especially in the case of the variables: "real GDP per capita" and "population density", but these values are not the "far outliers" type. The number of significant components was established by applying the following instruments: "eigenvalue-one criterion", the proportion in the total variation preserved by each component, and the "Scree Plot Test".

After the initial extraction of the components, their rotation was performed using the Varimax method to identify a simpler, easier to interpret structure of the found components, under the condition of maintaining a non-correlation of the factors. For a more detailed analysis of the nature and intensity of the interaction between the dependent variable (biocapacity reserve/deficit) and its determinants, the artificial components identified were included as explanatory variables in a multiple regression model, to study their impact on the biocapacity reserve/deficit. In applying the regression model,

the following steps were taken: The model was specified in the total population and in the sample, the model parameters were estimated, the validity of the model was tested, the quality of the model was evaluated, the statistical inference was applied to the model parameters, the intensity of the link between the variables was measured, and the assumptions specific to the simple linear regression model were verified.

## 4. Results

### 4.1. Description of the Ecological Footprint, Biocapacity, and Biocapacity Reserve/deficit of European Countries

Our humanity is at a crossroads, its unsustainable patterns of consumption and lifestyle contribute to the short-term and alarming destruction of everything that nature has created over millions of years. Over the last 10 years, the ecological footprint of the European Union has been on a more obvious downward trend, currently reaching 4.6 global hectares per person (2016), with the EU ranking second in the regions with the highest ecological footprint in the EU, after North America (6.6 global hectares per person, in 2016) (Figure 2). The EU biocapacity deficit peaked in the late 1970s (almost −4 global hectares per person in 1979) but has changed to −2.5 global hectares per person today, mainly by reducing the ecological footprint. With this value, the EU occupies the second position in the world, after North America, which registered a biocapacity deficit of −2.8 global hectares per person (Figure 3) [50]. If the lifestyle of a European resident were to be adopted, 2.8 planets would be needed to regenerate all the resources consumed, while at the world level, 1.7 planets would be needed.

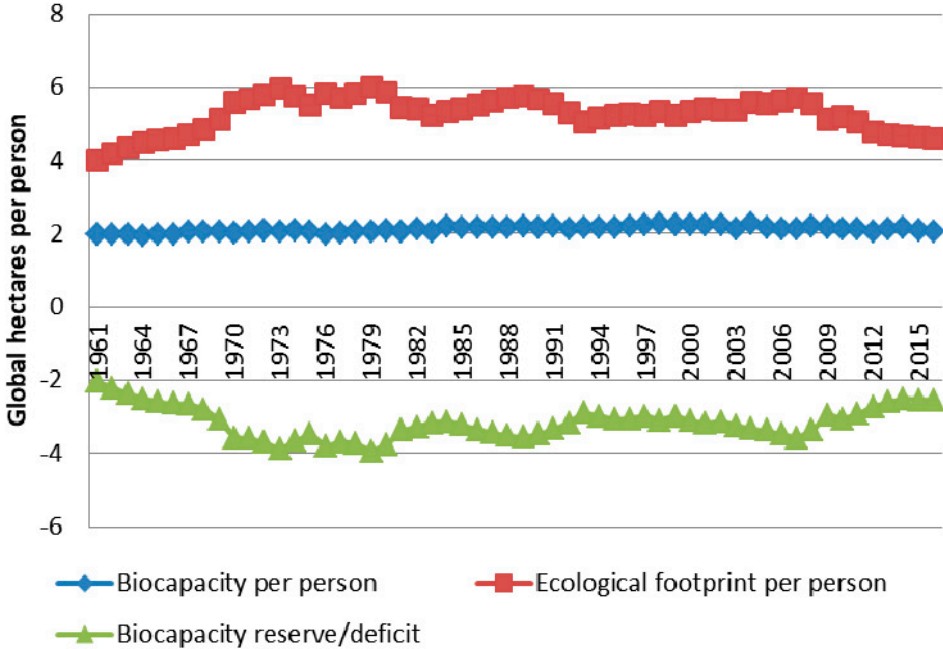

**Figure 2.** The evolution of the EU ecological footprint, biocapacity, and biocapacity reserve/deficit, between 1961–2016. Source: Authors' results, based on data provided by 2019 Edition National Footprint and Biocapacity Accounts.

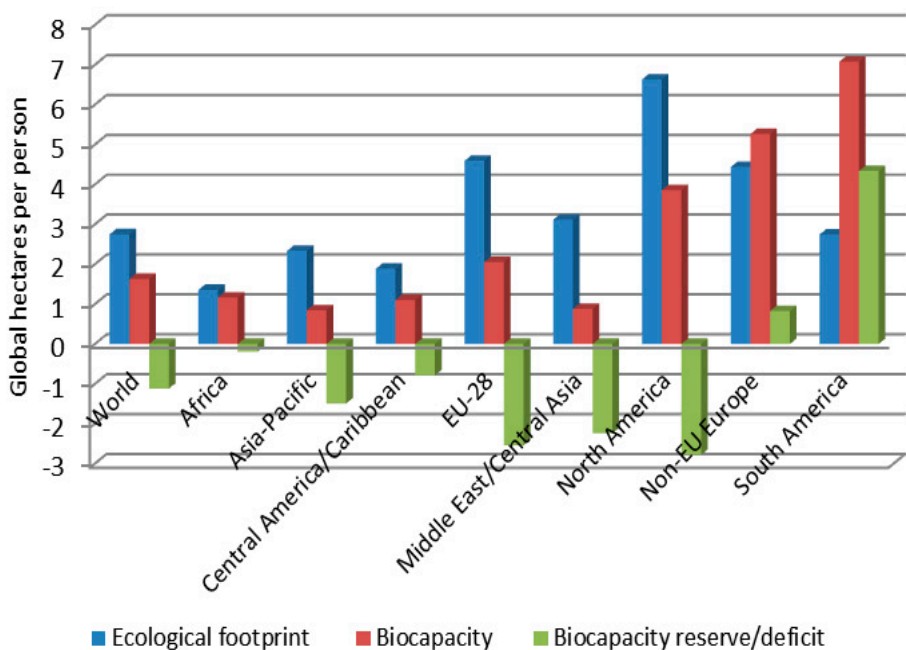

**Figure 3.** The level of ecological footprint, biocapacity, and biocapacity reserve/deficit on continents in 2016. Source: Authors' results, based on data provided by 2019 Edition National Footprint and Biocapacity Accounts.

Europe is largely dependent on the natural resources of other regions of the world, which exposes it to the instability in these areas. Measures taken to mitigate global warming and to stop environmental degradation could reduce Europe's fossil fuel imports, strengthening its independence [52].

Moreover, the planetary resources are unequally distributed and also unevenly used, so that, for example, European Union countries—which make up 7% of the world's population—use 20% of the planet's biocapacity.

If we analyze the distribution of the European countries according to the biocapacity reserve/deficit, one can notice a weak, negative skewness, with a low predominance of the high indicator values (Figure 4).

There were also some extreme values in the data series, generated by Luxembourg (with a biocapacity deficit of −11.67 global hectares per capita) and Finland (with a biocapacity reserve of 6.37 global hectares per capita). The median value of the biocapacity deficit was −2.65 global hectares per capita, with the series having a very high variability in territorial profile. It should be noted that 83% of the European countries included in the analysis had a biocapacity deficit, some of the countries with a high deficit (outside Luxembourg) being Belgium (with −5.47 global hectares per capita) and Malta (with −5.19 global hectares per capita). The top five countries with a positive biocapacity reserve are led by Finland (with 6.37 global hectares per capita), followed by Sweden (with 3.9 global hectares per capita), Norway, and the Baltic countries.

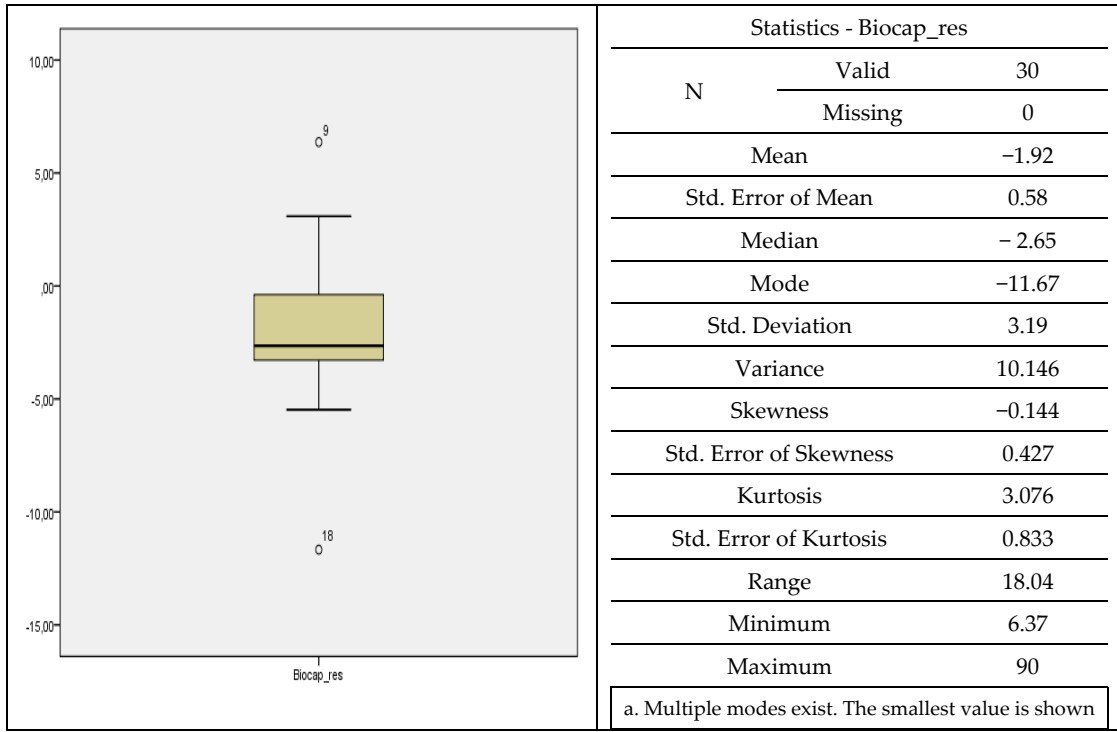

**Figure 4.** The results of the descriptive analysis of the biocapacity reserve/deficit of European countries in 2016; source: The authors' contribution, based on data provided by 2019 Edition National Footprint and Biocapacity Accounts.

*4.2. Identification and Description of the Behavioral Patterns of the European Countries According to the Biocapacity Reserve or Deficit and Its Determinants*

In order to identify and characterize the behavioral patterns of the European countries, Hierarchical Cluster Analysis was applied using the Ward method, which implies the cluster formation based on the criterion of minimizing the variability within groups. The aim of this analysis was to identify the homogeneous groups of European countries, which have similarities regarding the biocapacity reserve or deficit of, but also regarding its determinants in the economic, social, and environmental domains. Therefore, both the dependent variable and the 12 explanatory variables were considered as grouping criteria, their values—given the complexity and the variety of measurement units—were standardized. An optimal number of three clusters with similar size was identified. Thus, the first cluster includes 12 countries, the second cluster includes 9 countries, and the third includes 9 countries (Figure 5, Table 2).

**Table 2.** Cluster composition.

| Cluster | Countries |
|:---:|:---:|
| 1 | Austria, Belgium, Denmark, Finland, France, Germany, Ireland, Netherlands, Norway, Sweden, Switzerland, United Kingdom |
| 2 | Bulgaria, Croatia, Estonia, Hungary, Latvia, Lithuania, Poland, Romania, Slovakia |
| 3 | Cyprus, Czechia, Greece, Italy, Luxembourg, Malta, Portugal, Slovenia, Spain |

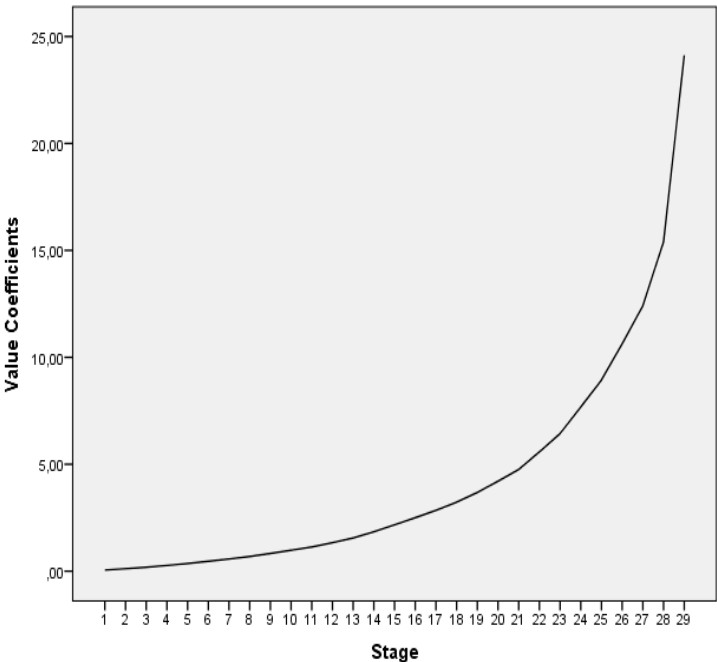

**Figure 5.** Agglomeration schedule—coefficients; source: Authors' contribution, based on data provided by 2019 Edition National Footprint and Biocapacity Accounts and EUROSTAT.

The Dendrogram show the cluster formation and composition (Figure 6).

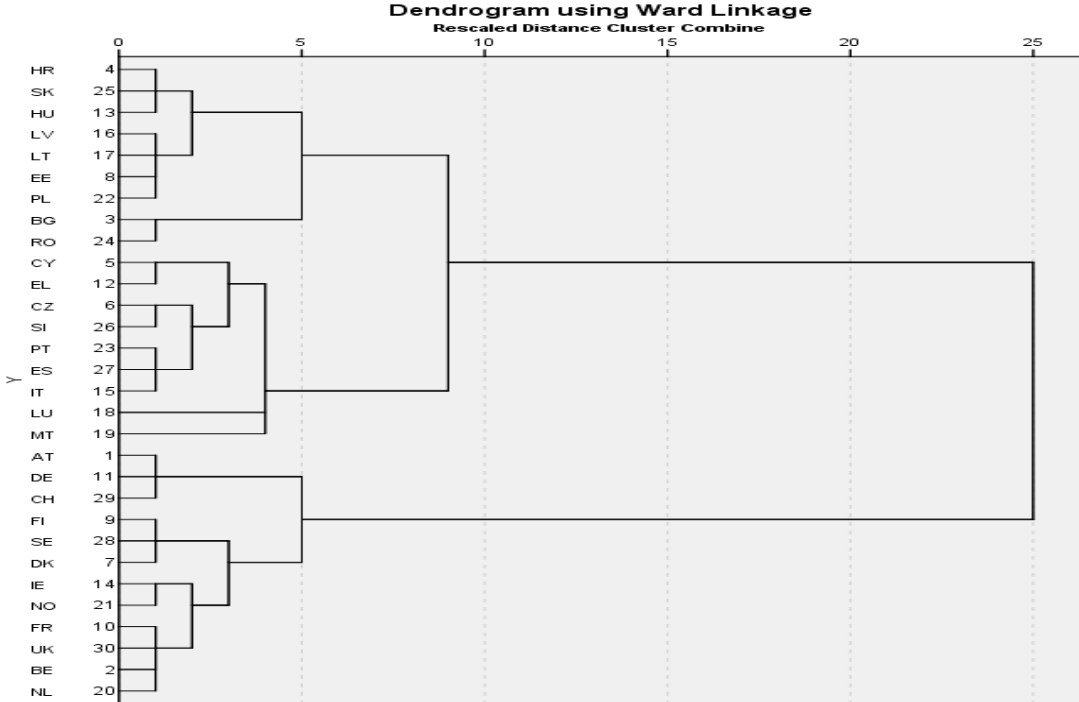

**Figure 6.** Dendrogram; source: Authors' contribution, based on data provided by 2019 Edition National Footprint and Biocapacity Accounts and EUROSTAT.

In order to compare the behavior of the countries in the three clusters, both central tendency and variability indicators of the clustering variables were taken into consideration, determining the average values and the standard deviations of the standardized hierarchical clustering variables. (Table 3, Figure 7.)

**Table 3.** The cluster situation of European countries according to the classification variables (z-scores).

| Cluster | Biocap_res | GDP_cap | Pop_dens | Life_exp | Health_exp | School_years | Part_educ | Underachv | RD_exp | Researchers | Mat_dep | Work_pov | Direct_loss |
|---|---|---|---|---|---|---|---|---|---|---|---|---|---|
| 1 | 0.13 | 0.75 | −0.01 | 0.57 | 0.93 | 0.58 | 0.87 | −0.59 | 0.96 | 0.95 | −0.67 | −0.57 | 0.43 |
| 2 | 0.54 | −0.93 | −0.38 | −0.87 | −0.97 | −0.39 | −0.57 | 0.42 | −0.81 | −0.89 | 0.83 | 0.43 | −0.77 |
| 3 | −0.72 | −0.08 | 0.39 | 0.61 | −0.26 | −0.39 | −0.59 | 0.36 | −0.47 | −0.37 | 0.06 | 0.32 | 0.19 |
| Legend: | | | | | | | | | | | | | |

| | |
|---|---|
| | the most unfavorable level |
| | average level |
| | the most favorable level |

Source: The authors' contribution, based on data provided by 2019 Edition National Footprint and Biocapacity Accounts and EUROSTAT.

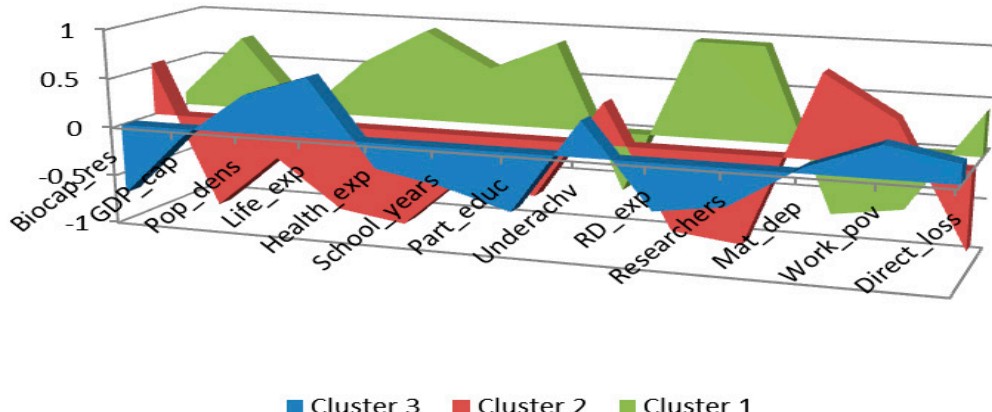

**Figure 7.** The situation of the clusters of European countries according to the classification variables (z-scores); source: Authors' contribution, based on data collected from 2019 Edition National Footprint and Biocapacity Accounts and EUROSTAT

Based on the results obtained by applying the Hierarchical Clustering Method, we can characterize the clusters of European countries, in terms of their behavior patterns regarding the classification variables.

The countries in the first cluster are characterized by the most favorable state of most variables. Thus, these countries present the highest level of economic development, reflected by a high per capita GDP level, well above average, fueled by a strong research-innovation sector, by the employed workforce and by the involvement of business sector, in conjunction with a high level of human resources development, in terms of health and education. At the same time, these countries are characterized by a high degree of social inclusion, with low material deprivation rates and in work poverty, but also by close-to-average levels of population density. Although the average biocapacity reserve of the countries in this cluster is negative, it hides the largest variations of all clusters: Here, North European countries, with a positive biocapacity reserve, coexist with highly developed countries in Central and Western Europe, with a biocapacity deficit; overall, the average biocapacity reserve is, however, slightly above the total average, with the largest economic losses due to extreme climatic phenomena in these countries. The extended and intense economic activity in these countries puts a great deal of pressure on the environment, but there are also sustained investments in environmental protection and innovation activity.

The countries in the second cluster are somewhat in opposition to those in the first cluster, with the most unfavorable aspects in most areas. This cluster includes countries in which the positive situation of the environmental variables (the highest average biocapacity reserve, the lowest economic losses under the action of the extreme phenomena generated by the climate changes) contrasts with the unfavorable situation in terms of the development of the economic activity and of the human resources (with the lowest GDP per capita, below average levels of population health and education, a high level of social exclusion, and low developed research innovation sector). This cluster includes the former socialist countries, the high biocapacity reserve, and the low economic losses caused by the extreme phenomena of countries in this cluster, compared to the countries in the other clusters being explained not by sustained investments in environment protection and restoration, but by a reduced and low-level economic activity, which does not put such a high pressure on nature.

The third cluster brings together the country group with high population density, with high life expectancy at birth, generally located in Southern Europe. The countries in this region, to which Luxembourg is added, have an average level of macroeconomic results and poverty, a development level below average of the research and innovation sector. At the same time, this cluster includes countries with a low and very low level of population education, from the perspective of the variables included in the study, although the differences between the averages of the three clusters are sensitive.

In terms of environmental variables, these countries have the largest biocapacity deficit, but also an average level of economic loss under the impact of extreme weather events. Luxembourg presents "outlier" values for some statistical variables: The highest GDP per capita (over $100,000 per capita) and the highest ecological footprint of consumption, as well as a reduced biocapacity, which led to the largest biocapacity deficit among all the countries included in the analysis (−11.67 global hectares per capita).

### 4.3. Principal Component Analysis

It is known that the variables "ecological footprint" and "biocapacity", based on which the biocapacity reserve or deficit is determined, refer to particularly complex concepts, their complexity being reflected in the very wide range of factors that may influence the changes of these variables. The Principal Component Analysis was applied in order to synthesize, to summarize the large-sized block of influence factors, extracting a limited number of independent dimensions, which incorporate a high variability of the dataset. Thus, the 12 variables selected and considered as determinants of the biocapacity reserve were included in the analysis, each with 30 values for the EU and non-EU countries. Following the analysis of the correlation matrix, the non-existence of very strong or very weak correlations between these variables resulted. The "Kaiser-Meyer-Olkin" indicator has a value of 0.733, suggesting—by exceeding the limit value of 0.6—the opportunity to apply the Principal Component Analysis method on the initial dataset. Following the application of the "Bartlett's Test of Sphericity", a significance level (Sig.) lower than the commonly used level of 5% leads to the rejection of the null hypothesis, regarding the existence of an identity matrix type correlation matrix, and therefore the Principal Component Analysis method can be applied (Table 4).

**Table 4.** KMO and Bartlett's Test.

| Indicators | | Values |
|---|---|---|
| Kaiser-Meyer-Olkin Measure of Sampling Adequacy. | | 0.733 |
| Bartlett's Test of Sphericity | Approx. Chi-Square | 254.910 |
| | df | 66 |
| | Sig. | 0.000 |

Source: The authors' contribution, based on data provided by 2019; Edition National Footprint and Biocapacity Accounts and EUROSTAT.

Three main components have been extracted from the initial set, as "Eigenvalues" for the first three components exceed 1.00 (5.83, 1.90, and 1.30—Table 5.). The first main component explains 48.58% of the variation of the data set, the second explains 15.84%, and the third explains 10.86%, with the three principal components together preserving 75.28% of the total variation of the dataset (Table 5, Figure 8).

**Table 5.** Total variance explained.

| Component | Initial Eigenvalues | | | Extraction Sums of Squared Loadings | | | Rotation Sums of Squared Loadings | | |
|---|---|---|---|---|---|---|---|---|---|
| | **Total** | **% of V** | **C %** | **Total** | **% of V** | **C %** | **Total** | **% of V** | **C %** |
| 1 | 5.830 | 48.583 | 48.583 | 5.830 | 48.583 | 48.583 | 3.827 | 31.892 | 31.892 |
| 2 | 1.901 | 15.839 | 64.422 | 1.901 | 15.839 | 64.422 | 3.793 | 31.612 | 63.504 |
| 3 | 1.303 | 10.862 | 75.284 | 1.303 | 10.862 | 75.284 | 1.414 | 11.780 | 75.284 |
| 4 | 0.800 | 6.665 | 81.949 | | | | | | |
| 5 | 0.703 | 5.858 | 87.807 | | | | | | |
| 6 | 0.507 | 4.225 | 92.032 | | | | | | |
| 7 | 0.332 | 2.766 | 94.798 | | | | | | |
| 8 | 0.203 | 1.693 | 96.490 | | | | | | |
| 9 | 0.150 | 1.246 | 97.736 | | | | | | |
| 10 | 0.131 | 1.092 | 98.829 | | | | | | |
| 11 | 0.081 | 0.675 | 99.503 | | | | | | |
| 12 | 0.060 | 0.497 | 100.000 | | | | | | |

Extraction Method: Principal Component Analysis. Note: V—Variance; C—Cumulative; Source: The authors' contribution, based on data provided by 2019 Edition National Footprint and Biocapacity; Accounts and EUROSTAT.

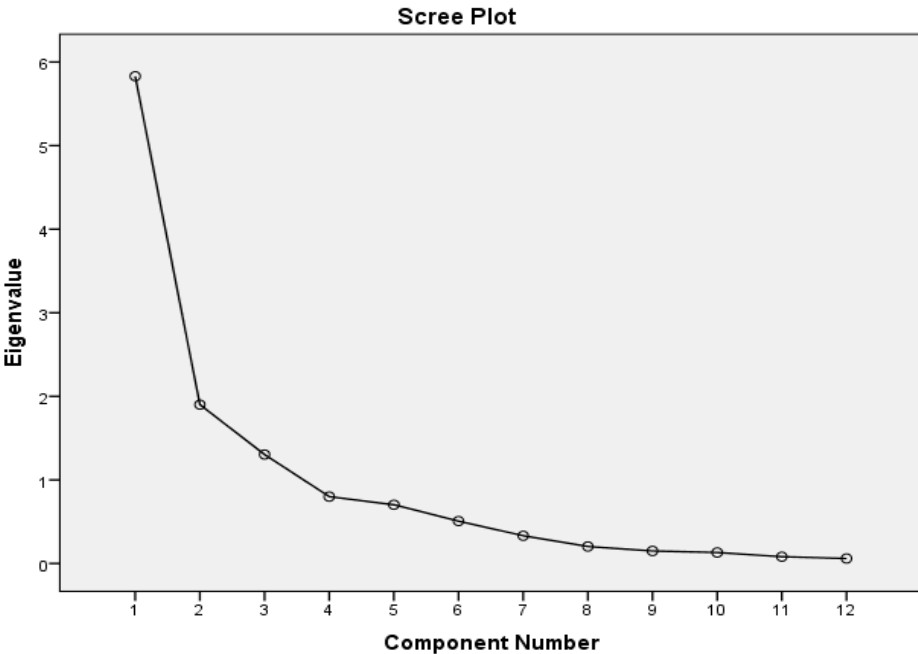

**Figure 8.** Scree plot. Source: The authors' contribution, based on data provided by 2019; Edition National Footprint and Biocapacity Accounts and EUROSTAT.

The three main components extracted have the following structure:

- Component 1 includes the variables: "Expected school years of pupils and students", "Participation rates in education", "Education underachieving level for 15- year-old students", "Researchers", "Material deprivation rate", "In-work at-risk-of-poverty rate". We assign this component the name of "Human development component" ("The education and social exclusion component").
- Component 2 includes the variables: "GDP per capita", "Life expectancy at birth", "Health care expenditures", "Business expense on R&D", "Direct losses due to extreme weather and climate related events". This component is named "The economic development, innovation, health, and environment component".
- Component 3 includes a single variable: "Population density" and can be called "The demographic component".

The correlation coefficients between each initial variable and the new main components identified are shown in Figure 9, which also contains the results of the factor rotation in order to construct simpler and more understandable structures of the components. The variables were standardized, as they presented complex and different measurement units, and there were some significant size differences between the values of the variables.

Thus, the variables that characterize the educational dimension ("*School_years*", "*Part_educ*") and the "*Researchers*" variable were strongly and positively correlated with Component 1; the variable "*Underachv*" and the variables that measure social inclusion ("*Mat_dep*", "*Work_pov*") were quite strongly and negatively correlated with Component 1. Variable "*GDP_cap*" and those that characterize the health domain ("*Life_exp*", "*Health_exp*"), "*RD_exp*", and "*Direct_loss*" were strongly and directly correlated with Component 2, while between the variable "*Pop_dens*" and Component 3 there was a very strong positive correlation (Table 6).

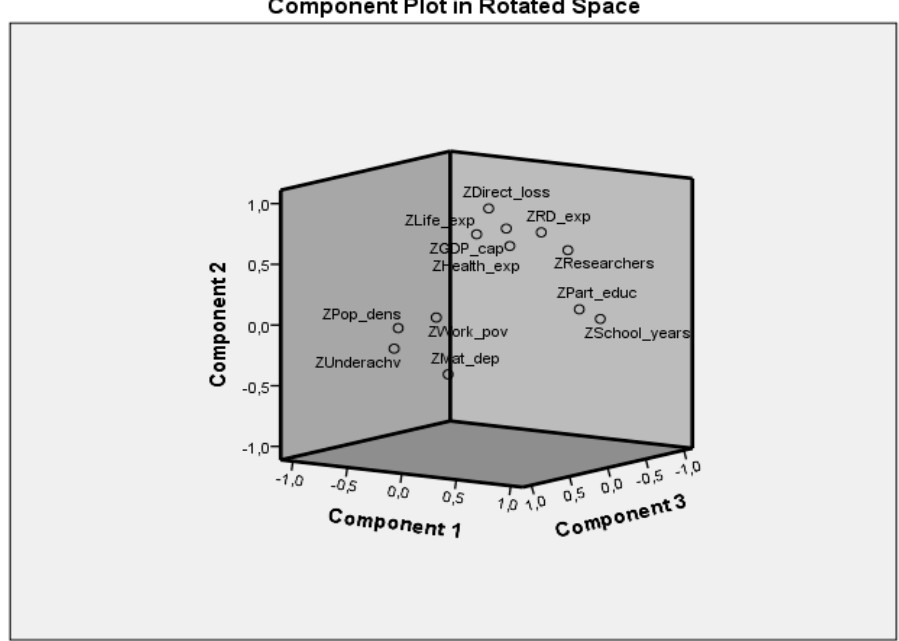

**Figure 9.** Component plot in rotated space. Source: The authors' contribution, based on data provided by 2019; Edition National Footprint and Biocapacity Accounts and EUROSTAT.

**Table 6.** Rotated component matrix [a].

|  | Component | | |
|---|---|---|---|
|  | **1** | **2** | **3** |
| Zscore (GDP_cap) | 0.222 | 0.777 | 0.059 |
| Zscore (Pop_dens) | −0.155 | 0.046 | 0.938 |
| Zscore (Life_exp) | 0.153 | 0.766 | 0.349 |
| Zscore (Health_exp) | 0.371 | 0.673 | 0.226 |
| Zscore (School_years) | 0.900 | 0.064 | −0.206 |
| Zscore (Part_educ) | 0.824 | 0.159 | −0.039 |
| Zscore (Underachv) | −0.765 | −0.303 | 0.119 |
| Zscore (RD_exp) | 0.445 | 0.748 | −0.081 |
| Zscore (Researchers) | 0.674 | 0.622 | −0.103 |
| Zscore (Mat_dep) | −0.576 | −0.559 | −0.318 |
| Zscore (Work_pov) | −0.721 | −0.113 | −0.370 |
| Zscore (Direct_loss) | −0.126 | 0.869 | −0.208 |

Extraction Method: Principal Component Analysis; Rotation Method: Varimax with Kaiser Normalization; a. Rotation converged in 5 iterations; Source: Made by the authors, based on data provided by 2019, Edition National Footprint and Biocapacity Accounts and EUROSTAT.

The communalities table (Table 7) shows the weight of the variation of each variable included in the analysis that is explained by the main components combined.

Thus, the highest values in this table show that—based on the combination of the main components—one can predict or explain 90.6% of the variation of the variable „Population density", 85.6% of the variation of the variable „Expected School Years", 85.1% of the variation of the variable „Researchers", and 81.4% of the variability of the variable „Impacts of extreme weather and climate related events". The components explain, in smaller proportions, the variation of the variables: „Health care expenditures" (64.2%), „GPD per capita" (65.7%), and „In-Work-at-risk of poverty rate" (66.9%).

Based on the coefficients included in the "Component Score Coefficient Matrix", the scores for each of the three components are determined, which will then be used in the regression analysis, in order to model the behavior of the biocapacity reserve/deficit.

**Table 7.** Communalities.

|  | Initial | Extraction |
|---|---|---|
| Zscore (GDP_cap) | 1.000 | 0.657 |
| Zscore (Pop_dens) | 1.000 | 0.906 |
| Zscore (Life_exp) | 1.000 | 0.732 |
| Zscore (Health_exp) | 1.000 | 0.642 |
| Zscore (School_years) | 1.000 | 0.856 |
| Zscore (Part_educ) | 1.000 | 0.705 |
| Zscore (Underachv) | 1.000 | 0.691 |
| Zscore (RD_exp) | 1.000 | 0.765 |
| Zscore (Researchers) | 1.000 | 0.851 |
| Zscore (Mat_dep) | 1.000 | 0.745 |
| Zscore (Work_pov) | 1.000 | 0.669 |
| Zscore (Direct_loss) | 1.000 | 0.814 |

Extraction Method: Principal Component Analysis; source: Made by the authors, based on data provided by 2019 Edition National Footprint and Biocapacity Accounts and EUROSTAT.

### 4.4. Multiple Linear Regression

In this stage of the analysis, the existence, form, and intensity of a relationship between the biocapacity reserve/deficit and the artificial main components identified following the application of the Principal Component Analysis method were analyzed in detail. Specifically, the behavior of the biocapacity reserve/deficit was modeled according to these main components, with the nature and direction of the influence of the components on the biocapacity reserve being analyzed, using a multifactorial linear regression model. The three main components (Component 1—"Education and social exclusion component", Component 2—"Economic development, innovation, health, and environment component", and Component 3—"Demographic component") were included as explanatory variables in the model, and biocapacity reserve/deficit represents the explained variable. The model has the following general form:

$$Biocap\_res_i = \beta_0 + \sum_{j=1}^{k} \beta_j \cdot Comp_j^i + \varepsilon_i i = \overline{1,N} \tag{1}$$

where

$\beta_j$, $(j = \overline{0,k})$ represents the regression model parameters ($\beta_0$ is the intercept parameter, and $\beta_j$, $j = \overline{1,k}$ is the slope parameter);

$k$ represents the number of main components (explanatory variables);

$\varepsilon_j$, $(j = \overline{1,k})$ represents the error random term, which synthesizes the influence of random or residual factors;

$N$ is the total population size (number of countries);

$Comp_j^i$ is the value (score) of component j at the statistical unit (country) i;

$Biocap\_res_i$ is the value of the variable Biocapacity reserve/deficit for country i.

Following the data processing by applying the multiple linear regression method, we can say that the model explained 48.5% of the variation of the biocapacity reserve, with a statistically significant explanatory power over the dependent variable, and the relationship between the four variables had a high intensity (correlation ratio = 0.697) (Table 8)

**Table 8.** Model summary [b].

| Model | R | R Square | Adjusted R Square | Std. Error of the Estimate | Durbin-Watson |
|---|---|---|---|---|---|
| 1 | 0.697 [a] | 0.485 | 0.426 | 2.41332 | 2.283 |

a. Predictors: (Constant), $Comp_1$, $Comp_2$, $Comp_3$; b. Dependent Variable: Biocap_res; Source: made by the authors, based on data provided by 2019 Edition National Footprint and Biocapacity Accounts and EUROSTAT.

By applying the analysis of variance method (ANOVA) and Fisher's test, the hypothesis of the validity of the regression model was tested, a hypothesis that can be accepted for a 95% confidence level (Sig. = 0.001 < 0.05) (Table 9)

**Table 9.** ANOVA results [a].

| Model | | Sum of Squares | df | Mean Square | F | Sig. |
|---|---|---|---|---|---|---|
| | Regression | 142.805 | 3 | 47.602 | 8.173 | 0.001 [b] |
| 1 | Residual | 151.427 | 26 | 5.824 | | |
| | Total | 294.231 | 29 | | | |

b. Predictors: (Constant), $Comp_1$, $Comp_2$, $Comp_3$; a. Dependent Variable: Biocap_res; source: Made by the authors, based on data taken from 2019 Edition National Footprint and Biocapacity Accounts and EUROSTAT.

The obtained estimators of the regression model parameters are: $b_0 = -1.92$; $b_1 = 1.462$, $b_2 = -1.312$, $b_3 = -1.032$, with the model having the following form (Table 10):

$$Biocap\_res_i = -1.92 + 1.462 \times Comp_1^i - 1.312 \times Comp_2^i - 1.032 \times Comp_3^i + \varepsilon_i i = \overline{1,30} \qquad (2)$$

**Table 10.** Model coefficients.

| Model | | Unstandardized Coefficients | | Standardized Coefficients | t | Sig. | 95,0% Confidence Interval for B | | Collinearity Statistics | |
|---|---|---|---|---|---|---|---|---|---|---|
| | | B | Std. Error | Beta | | | Lower Bound | Upper Bound | Tolerance | VIF |
| 1 | (Constant) | −1.920 | 0.441 | | −4.358 | 0.000 | −2.826 | −1.015 | | |
| | $Comp_1$ | 1.462 | 0.448 | 0.459 | 3.263 | 0.003 | 0.541 | 2.384 | 1.000 | 1.000 |
| | $Comp_2$ | −1.312 | 0.448 | −0.412 | −2.928 | 0.007 | −2.233 | −0.391 | 1.000 | 1.000 |
| | $Comp_3$ | −1.032 | 0.448 | −0.324 | −2.302 | 0.030 | −1.953 | −0.111 | 1.000 | 1.000 |

Following the application of "t" test, the statistical significance of the four model parameters was demonstrated, obtaining minimum significance levels below the threshold of 0.05 (Sig.1 = 0.000, Sig.2 = 0.003, Sig.3 = 0.007, Sig.4 = 0.03).

The study continues by analyzing the sign and value of the estimators for each parameter of the regression model. The estimator of Component 1 ("Education and social exclusion component") was positive, which indicates a positive, direct correlation between this main component and the biocapacity reserve/deficit. In other words, it is estimated that an increase in the score related to the "Education and social exclusion component" will lead to an increase in the biocapacity reserve. Component 2 ("Economic development, innovation, health, and environment component") negatively influenced the variation of the biocapacity reserve (negative coefficient), which means that an increase in the score related to this component will lead to the reduction of the biocapacity reserve and to an increase in the biocapacity deficit. Component 3 ("Demographic component") also had an inverse influence on the variation of the biocapacity reserve (negative coefficient), an increase of the score related to this component leading to a decrease of the biocapacity reserve. In order to assess the quality of the linear regression model, the following hypotheses were tested: The linear functional form, the normality of the residuals' distribution, the residuals' homoscedasticity, the non-correlation of the explanatory variables with the residual ones, the errors' non-autocorrelation, and the non-correlation of the exogenous variables. Following the application of the statistical tests, these hypotheses were validated (Figures 10 and 11).

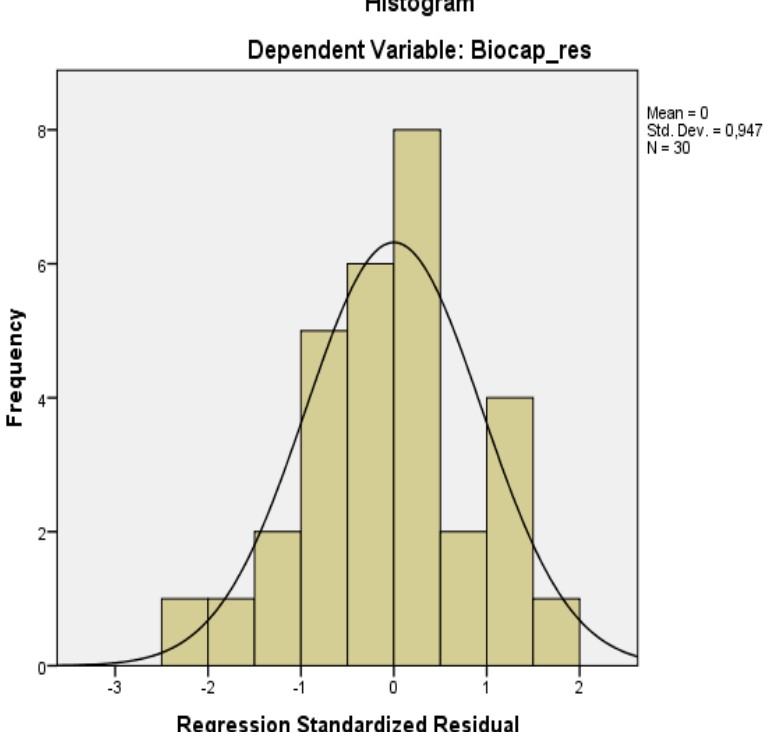

**Figure 10.** Standardized residuals histogram. Source: Authors' results, based on data provided by 2019 Edition National Footprint and Biocapacity Accounts and EUROSTAT.

**Normal P-P Plot of Regression Standardized Residual**

**Figure 11.** Normal P-P plot of regression standardized residuals. Source: Authors' results, based on data provided by 2019 Edition National Footprint and Biocapacity Accounts and EUROSTAT.

In conclusion, following the application of the multidimensional statistical analysis, the four hypotheses previously formulated referring to the categories of determinants that can significantly influence the behavior of the biocapacity reserve/deficit have been validated

## 5. Discussion

The identification of the main determinants whose significant influence can explain the variations in biocapacity reserve/deficit is important for establishing the intervention tools on the optimization of the biocapacity deficit, in the economic, social, and environmental field.

The analysis shows that a potential increase in biocapacity reserve/deficit is the result of the growth of "*School_years*", "*Part_educ*", and "*Researchers*" on the one hand and the reduction of "*Underachv*", "*Mat_dep*", and "*Work_pov*" on the other, all being elements of Component 1, an improved score of "Education and social exclusion" Component.

The explanation of these influences lies in the fact that a higher educational level and higher knowledge accumulations lead to a better professional training of the labor force, which, together with the reduction of the poverty rate, of social exclusion, represent the basis for an increase in the living standard and life quality. Thus, the premises are created for a more efficient use of material and human resources, for obtaining higher results with a lower resource consumption, which can lead to the reduction of the ecological footprint and to an increase of the biocapacity reserve. At the same time, an increase in the share of employed people in the research innovation domain, an area that requires a higher skilled workforce, leads to the possibility of obtaining new technologies and less polluting materials, to reduce the exploitation of non-renewable natural resources, to appeal, to a large extent, at alternative energy sources (green energy, from renewable sources), determining a decrease in the pressure of human activity on the environment and a restoration of the biocapacity reserve. The results are not entirely consistent with those of other studies, still they could be explained by the fact that the applied method did not analyze the influence of each variable, taken separately, on the biocapacity reserve, but of their interaction, in a complex component.

The influence of Component 2 ("Economic development, innovation, health, and environment component") on the variation of the biocapacity reserve was inverse, revealing that an increase in the score for Component 2 leads to a reduction in the biocapacity reserve. Indeed, countries with higher macroeconomic outcomes are also highly polluting and impose greater pressure on the environment, leading to a deterioration of its quality, but also to negative environmental reactions on people. Thus, the manifestation of extreme phenomena is extended, leading to substantial economic losses, all of which result in the reduction of the biocapacity reserve. The results are similar to those obtained by Jia et al. (2009) [13] and Tang, Zhong, and Liu, S. (2011) [14], Jaffe, Newell, and Stavins (2002) [22], and Bel and Joseph (2018) [23] with respect to the economic development level and the research innovation activity and confirm the positive correlation between the population health state and the ecological footprint. Component 3 ("Demographic component") negatively influenced the behavior of the biocapacity reserve/deficit. Indeed, an increase in the score for Component 3 implies an increase in population density, which means a higher consumption of goods and services, an additional pressure on the environment and a higher risk of deterioration, as a result of which the biocapacity reserve may be diminished. The results are in accordance with those of Jia et al. (2009) [13] and Tang, Zhong, and Liu, S. (2011) [14], which identified the demographic factor as a determinant with a significant direct action on the ecological footprint, and therefore with the effect of lowering the biocapacity reserve.

The study is limited to European countries. Thirty European countries (28 EU member countries, Switzerland, and Norway) were included in the analysis, excluding countries for which there was a statistical information deficit. The authors intend to continue the research on the behavior of the biocapacity reserve or deficit by identifying other factors with significant influence, in order to highlight new ways to optimize its level, continuing the sustainable path of economic development. At the same time, the authors intend to extend the study to other non-European countries, with increased ecological footprint level and a large biocapacity deficit.

The research carried out focuses on a response variable less approached from an econometric perspective in the specialized studies: The biocapacity reserve or deficit, which integrates the double plan of the resources consumption and their regeneration capacity. The originality note of the article is also given by the selection of the determinants of the biocapacity reserve/deficit, factors that

cover all the three main defining dimensions of sustainable development: The economic, social, and environmental dimension. The article focuses on the social component, since man is the link that joins the economy and the environment. In this respect, there were selected variables that allow a more detailed characterization of the social dimension—from an educational, health, and social exclusion perspective. The novelty also consists in identifying the behavioral patterns of the European countries from a double point of view: That of biocapacity reserve/deficit, and that of its determinants, allowing the formulation of a set of recommendations of economic, social, and environment measures for optimizing the biocapacity reserve, which must become one of Europe's priorities.

The influence of the regression model components on the behavior of biodiversity is to some extent consistent with the research results of Lazarus et al. [54], Lenzen et al. [55], as well as that of Rudolph and Figge [56]. They show the influence of the economic and political globalization policy on the behavior of ecological footprint and implicitly of the biocapacity, in the short term. At the same time, the results obtained are similar to those of Terziovski and Guerrero [57], which show on the one hand a positive impact of sustainable practices, and on the other hand the importance of innovation performance, although both positive and negative influences on the environment, under the impact of technology, appear over time.

The results of the article indicate significant differences between the biodiversity level in the developed European countries, compared to those of the developing countries, as in Lenzen et al. [55], who reached similar conclusions, but for a larger group of countries.

The conclusion is that all countries should strive to evolve favorably in the context of biodiversity behavior that supports the development of the global sustainability process.

## 6. Conclusions

The research carried out by scientists and the results of their study highlighted the risk of an ecological collapse, which could be extended on entire planet. All countries must act together to stop the danger of the extinction of many species and of global warming, allowing the planet a respite moment, to restore its own resources; otherwise, it is questioning the very existence of mankind.

Based on this aspect, the present research has focused on analyzing the fluctuations of the biocapacity reserve/deficit, in relation to three main complex components (the component of education and social exclusion, the component of economic development, innovation, health, and environment, and the demographic component). They were identified following the application of the Principal Component Analysis, taking into account 12 variables with significant influence on biocapacity behavior, as well as by clustering 30 EU and non-EU countries.

All the three main components have a significant influence and a significant explanatory power on the variability of the biocapacity reserve or deficit and can constitute important tools in adopting policy measures aiming at its optimization.

Each country can use its own material and human resources, its own knowhow, to frame the results of its economic activity on a sustainable path. An advantage of the European countries in the first cluster—in addition to the highest macroeconomic results—is the high level of human resources development, with the highest percentages of GDP allocated to health, the largest participation in the education process and with high school performance. Thus, it is recommended for the first cluster countries to use their investments in the research-development-innovation sector, (based on which they have obtained a reduced material consumption and a high resource productivity) for reducing the ecological footprint of consumption, for strengthening the biocapacity reserve, to encourage the inhabitants in adopting sustainable consumption models, based on environmental responsibility. This will ensure a reduction of pollution and—in general—a reduction of the negative impact of human activities on the environment, accompanied by a reduction of the economic losses caused by the manifestations of extreme phenomena.

According to the results obtained, the application of environmental protection measures is important to promote biodiversity in the context of sustainability. Thus, a strong orientation towards

a new vision of human relationship with nature is required to ensure and support ethical behavior towards it. The priority direction is to support the development of a harmonious human-environment relationship based on the ethics of human-nature relationships [58].

The countries in the second cluster, characterized by a low output level of the macroeconomic activity, are recommended to improve these results in intensive, qualitative ways, respectively by increasing the quality of the human resource and the efficiency of the natural resources use, by optimizing the general resource productivity. This can be achieved as follows: In the educational field—by reducing the early school leaving in some countries, by increasing the enrollment rate in education, by encouraging high school performance; in the health domain—through more substantial investments in the health sector and increasing the accessibility of the population to the medical services; in the development research field—by attracting public and/or private resources necessary for the development of the activity, by motivating, stimulating the employment in this sector, and by using the results of the research innovation in creating eco-friendly technologies, which will protect and raise the environment quality. The innovative factor is essential both in supporting sustainable economic development and in modeling biodiversity. The focus of the research activities on increasing the resources productivity and ecological investments will lead to positive evolutions of biodiversity. The results aim at obtaining advanced technologies based on the most efficient use of non-renewable resources, as well as on increasing the use of renewable energy. [17,59]. In this direction, it is very important for the government and other competent institutions of the countries to support the research and development activity by providing tax incentives or funds. All these approaches are directed towards obtaining and applying innovative results in greening the production process, with a direct impact on biocapacity. The countries in the last cluster have the highest level of both the biocapacity deficit and the economic losses under the impact of extreme weather events. Measures are needed to promote sustainability, to make consumption more responsible, to focus on "green" models and products, with a reduced negative impact on the environment. The situation of the education sector, from the perspective of the selected variables is similar to that of the countries in the previous cluster, so the measures recommended to them are also welcomed within the third cluster. Material and human resources must be used in such a way as to reduce the negative reaction of the damaged environment on human communities and to reduce the economic losses caused by this reaction [60,61].

In the globalization context, the European countries must help each other in softening the imbalances between human development and the environment state, for the benefit of all its inhabitants. Seventy-five percent of the inhabitants of Europe would like the European Union to act more towards environmental protection—this places third in the top priorities [62].

It is estimated that following the adoption of measures to sustain the production, consumption, and use of energy, following the implementation of the action plan established in 2015, in Paris, on climate change, 2 million new jobs could be created in Europe, and the circular economy could generate total benefits of € 1.8 trillion by 2030 [63]. Thus, joint actions of decision-makers from all European countries will have the effect of increasing security, improving population health, increasing the number of quality jobs, and Europe's competitiveness.

Humanity is at the crossroads between the triumph of life and its failure and it is up to each decision maker, to each of us to choose the right path, which leads to the planet's rescue and to our salvation.

The study is limited to European countries. Thirty European countries (28 EU member countries, Switzerland, and Norway) were included in the analysis, excluding countries for which there was a statistical information deficit. The authors intend to continue the research on the behavior of the biocapacity reserve or deficit by identifying other factors with significant influence, in order to highlight new ways to optimize its level, continuing the sustainable path of economic development. At the same time, the authors intend to extend the study to other non-European countries, with increased ecological footprint level and a large biocapacity deficit.

**Author Contributions:** In this article, all the authors were equally involved in the documentation phase, in choosing the research methodology, in data analysis, as well as in results analysis and in discussions. All the authors have equally participated to the manuscript preparation and have approved the submitted manuscript. All authors have read and agree to the published version of the manuscript.

**Funding:** This research received no external funding.

**Conflicts of Interest:** The authors declare no conflict of interest.

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
