# Peer review of "Biocapacity—Premise of Sustainable Development in the European Space"

_sustainability, doi:10.3390/su12031037_

Round 1

Reviewer 1 Report

This paper is interesting which posed a conception of biocapacity deficit and reserve as a way to understand the relations between human development and environmental conservation. However, the descriptions throughout the paper are wordy and repeated. The structure of the paper is not straightforward to me. I am not so clear what the authors were trying to do and what the key messages delivered in the study. 

The background introduction and literature review are poorly developed. The discussions are very wordy, rather than sticking to the key point of the argument. Duplications appear in many places at times. For example, the beginning of the Introduction and the literature review seem to be the same things. I cannot see the need of the study presented after reading these sections. As well, the review should be more criticised than simply listing what has been done by whom. The authors need to tighten the discussions with clearer logic links. Research objectives are described both in the introduction and methodology which are different. What are the real objectives? Also, in the introduction, one of the authors’ goals is to “identify main determinants of biocapacity reserve/deficit”. However, in the beginning of the methodology, the authors state that according to the main determinants. I am lost with the wordings and logic linkage.

It’s not clear to me that why the seven dimensions are the case for the biocapacity. What are differences and what the interconnections among them?

The mothedology section is not clearly described. A series of analyses seem to be involved. However, it’s hard to follow with the wordings throughout. It would be helpful for readers to understand the methodology with a diagram showing the conceptual framework of the study.

Reviewer 2 Report

Thank you for the opportunity to review this interesting paper, which tackles an important issue, albeit one which has been addressed through a number of methodologies in the past (planetary boundaries, global capacity etc).

I have some suggestions which I feel could improve your paper.

Overall the English is good, but there is a lot of use of poor syntax, superfluous use of the particle 'the', and use of non-traditional quotation marks, these should all be superscript. "like this". The referencing needs to be improved, some of the references come too late in the prose, i.e. your use of the term 'earth overshoot day' is unfamiliar and should be referenced immediately. This should be maintained for any evidence which is 'known as' or 'stated by' etc. The literature review requires some sub headings, as it is difficult to follow the logic and it is a bit scattered at the moment. In the methods section, you need to provide a justification for why you chose the seven dimensions, and which you regress using specific indicators, otherwise it is a subjective exercise. Also in the methodology, you do not need to tell us that you used SPSS - this is not relevant to the research. In the results, your use of positive and negative numbers is problematic. Please rewrite the biocapacity section so that readers can more easily follow it. Also in results, please make some linkages to other research in this area, how do their results compare with yours? Also, the clustering is explained very quickly, and difficult to follow - some additional work here could improve this. In the discussion, I find that the decrease in material deprivation rate being linked with improved biocapacity outcomes is counter-intuitive.  I think you will find that most scholars agree that the improvement of development level, or quality of life in a nation actually leads to a burst of increased consumption, and higher pressure on biocapacity reserves, at least in the development stage - please describe this interesting result - is it an artifact of your methodology, or does it assume that developing nations will immediately achieve the efficiency gains of their developed peers? The conclusions rehash the results - avoid this. Please use this section to provide punchy, relevant implications, including those on national or regional policy if possible. The results sections should stand on it's own and not need further description in the conclusions - some discussion of results in the discussion section is fine.

Round 2

Reviewer 1 Report

Thanks for sending back the revised manuscript which has addressed to some degree the previous comments. However, I cannot change my recommendation on this paper. There is very few contribution to the field. The conception of biocapacity is just a term that is used to deal with the relation between human and nature. In fact, the conception is very similar to many others such as environmental carrying capacity. The authors failed to engage with those cognate conceptions. Thus, it is difficult to see the contribution of the work presented.

Reviewer 2 Report

I am satisfied that the authors have made the appropriate changes required for acceptance of this article. Please make one small change: You use the term 'the last one' in line 81 of the lit review - please change this to 'the latter'.